# The Bone Buttress Theory: The Effect of the Mechanical Loading of Bone on the Osseointegration of Dental Implants

**DOI:** 10.3390/biology10010012

**Published:** 2020-12-28

**Authors:** David Chavarri-Prado, Aritza Brizuela-Velasco, Ángel Álvarez-Arenal, Markel Dieguez-Pereira, Esteban Pérez-Pevida, Iratxe Viteri-Agustín, Alejandro Estrada-Martínez

**Affiliations:** 1Department of Surgery and Surgical Specialties, School of Medicine and Health Sciences, University of Oviedo, 33006 Oviedo, Spain; brizuela@uniovi.es (A.B.-V.); arenal@uniovi.es (Á.Á.-A.); UO254090@uniovi.es (M.D.-P.); UO261326@uniovi.es (A.E.-M.); 2Department of Surgery, Faculty of Medicine, University of Salamanca, 37007 Salamanca, Spain; eperezpevida@usal.es; 3Faculty of Health Sciences, Miguel de Cervantes European University, 47012 Valladolid, Spain; 4Department of Pharmacology and Physiology, School of Medicine, University of Zaragoza, 50009 Zaragoza, Spain; 731318@unizar.es

**Keywords:** implant stability, resonance frequency analysis, osseointegration, immediate loading, BIC

## Abstract

**Simple Summary:**

The bone, as a vertebrate support tissue, is capable of adapting its structure and function to the mechanical demands resulting from the loads that are produced during the performance of its activity. This regulatory action also occurs during the healing processes of a fracture. The purpose of this study was to determine to what extent a dynamic load was capable of modulating the bone healing response around a titanium implant. The study was carried out on experimental rabbits, to which dental implants were placed in the tibiae and there were two test groups, one in which they did not undergo exercise during healing period and another that ran daily during this process on a treadmill. The trail results showed an improvement in the osseointegration process of the implant in the group in which it was subjected to load. The importance of these results is that it opens the door to a better understanding of the mechanisms that can modulate bone healing, especially around dental implants, supporting implant loading protocols that are based on efficiency.

**Abstract:**

Objectives: To determine the effect of mechanical loading of bone on the stability and histomorphometric variables of the osseointegration of dental implants using an experimental test in an animal model. Materials and Methods: A total of 4 human implants were placed in both tibiae of 10 New Zealand rabbits (*n* = 40). A 6-week osseointegration was considered, and the rabbits were randomly assigned to two groups: Group A (Test group) included 5 rabbits that ran on a treadmill for 20 min daily during the osseointegration period; Group B (Controls) included the other 5 that were housed conventionally. The monitored variables were related to the primary and secondary stability of the dental implants (implant stability quotient—ISQ), vertical bone growth, bone to implant contact (BIC), area of regenerated bone and the percentage of immature matrix. Results: The results of the study show a greater vertical bone growth (Group A 1.26 ± 0.48 mm, Group B 0.32 ± 0.47 mm, *p* < 0.001), higher ISQ values (Group A 11.25 ± 6.10 ISQ, 15.73%; Group B 5.80 ± 5.97 ISQ, 7.99%, *p* = 0.006) and a higher BIC (Group A 19.37%, Group B 23.60%, *p* = 0.0058) for implants in the test group, with statistically significant differences. A higher percentage of immature bone matrix was observed for implants in the control group (20.68 ± 9.53) than those in the test group (15.38 ± 8.84) (*p* = 0.108). A larger area of regenerated bone was also observed for the test implants (Group A 280.50 ± 125.40 mm^2^, Group B 228.00 ± 141.40 mm^2^), but it was not statistically significant (*p* = 0.121). Conclusions: The mechanical loading of bone improves the stability and the histomorphometric variables of the osseointegration of dental implants.

## 1. Introduction

The osseointegration of dental implants has often been described as a model of primary or direct healing of a bone fracture. Mechanical stability and a favourable biological environment are essential for the success of both processes [1].

Mechanical stability refers to the fact that an excessive force at the fracture edges or between the osteoid apposition line and the titanium surface during osseointegration of a dental implant can lead to successive ruptures of the newformed capillaries and impair the repair process mediated by fibrous tissue [2].

A study recently published by the same research group that conducted this study showed that the use of integrated dental implants and those that were immediately loaded after surgical placement through masticatory loading increased their stability [3]. These results suggest that positive functional results can be expected during the integration of implants. Several theoretical studies related to traumatology report that dynamic loading during the bone healing process can increase the levels of mesenchymal stem cells (MSC) in the endosteum, chondrogenic growth factors in the periosteum and cortical areas around the callus of the fracture [4].

There is sufficient scientific evidence in the literature to support the success and survival of immediately loaded dental implants, including single implants that do not have the mechanical advantage of immobilisation and immediate implants in post-extraction sites that have their lateral stability compromised by a gap between them and the post-extraction alveolar socket [5,6,7].

However, successful osseointegration is not enough, since several studies have already shown that bone formation can be stimulated when certain forces are applied to the implant during integration. This leads to an increase in the bone-to-implant contact (BIC) ratio or the amount of mineralised bone at the bone-to-implant interface [8,9,10,11,12]. Moreover, immediate loading seems to increase alveolar bone neoformation on the surface of the implants [12]. Although the underlying cellular or molecular mechanisms are still not clear, some studies indicate that there is a dose-dependent relationship between the magnitude of the applied force and the production and rearrangement of mineralised material and bone tissue [13].

Finally, the healing process is complex and multifaceted. The biomechanical stimulus, consisting of the application of a force to the bone, can increase the density, size and interconnectivity of the bone trabeculae. Most importantly, it can significantly alter their three-dimensional structure, and they can assume an arrangement that is functionally oriented to the direction of the received load [14,15,16]. In this regard, Lanyon has already demonstrated that the main orientation of the bone trabeculae coincides with the main directions of the force, which could be traction or compression, to which the bone is subjected [17,18]. Hayes and Snyder also demonstrated through a finite element analysis that the orientation of the bone trabeculae is strongly correlated with the main directions of the force that the bone receives [19]. Both studies support the theory of the alignment of the bone trabeculae with the main directions of the force proposed by Wolff in the 19th century [20,21]; the theory can be explained by the concept of mechanotransduction. The osteocytes act as pressure sensors, and they sense the alterations in the load received by the bone. Once the mechanical stimulus is sensed, the osteocytes transform this corresponding signal into an intracellular biochemical response and transmit it to the osteoblasts and osteoclasts. The osteoblasts and osteoclasts, in turn, produce or eliminate bone, respectively, depending on the demand [22]. The transmission of this information involves a multitude of autocrine and paracrine factors, but the information is transmitted through a dendritic network that is similar to the neural network of the brain [23]. Since the trabecular bone is an orthotropic material, the elastic energy applied to the material by an arbitrary force minimises when it is aligned with the main direction of the applied force [24]. The load acting during the early phase of osseointegration increases the production of transverse collagen fibres and the mineral density. The studies by Traini et al. showed a greater organisation of collagen fibres around loaded implants as well as a higher number of transverse fibres perpendicular to the implant axis [25,26,27]. This demonstrates that the immediate loading protocol is beneficial; loaded implants have a better organisation of the peri-implant bone than unloaded implants. The authors state that the quantity and orientation of the collagen fibres surrounding the implant can serve as a reliable measure of the quality of osseointegration [26,27].

In clinical practice, this can mean that immediate or early loading protocols in oral implantology are not only feasible and predictable [10,11,12], but they can also be favourable and beneficial for the arrangement and structuring of the bone around the implant depending on the load [9,13,14,15,16]. As a result, conventional loading protocols -in which the implant is restored a few months after placement, once it is osseointegrated, can condemn the bone to suffer a phenomenon of later bone remodelling, to redirect the bone in relation to the received load relying on inflammatory proceses, that not always will be respectful with the maintenance of the peri implant marginal bone [3].

However, there is currently insufficient scientific evidence on the effect of a biomechanical stimulus on the peri-implant bone during osseointegration. For this reason, the general objective of our study was to determine the outcome of mechanical loading after implant placement in an experimental animal model using histological and histomorphometric variables. The specific objectives were to analyse the effect of mechanical loading on the biological stability of implants, determine the effect of mechanical loading on the histological characteristics of the newformed peri-implant bone and the bone-to-implant contact ratio, and compare the patterns that emerge in the three-dimensional arrangement of peri-implant bone tissue according to the established loading protocol.

## 2. Material & Methods

The experiment focused on a rabbit tibia study that involved implant placement to compare the arrangement and characteristics of peri-implant bone tissue formed during osseointegration in animals that had and had not received biomechanical stimuli.

### 2.1. Reporting of In Vivo Experiments

This study used the guidelines of “Animals in Research: Reporting In Vivo Experiments” (ARRIVE) for reporting animal in vivo experiments to facilitate the reproducibility and adequate communication of the findings [28].

### 2.2. Specimens

This study involved 10 mature male New Zealand Rabbits, with each weighing 3.5–4 kg. Ethical approval was obtained from the Ethics Committee for Animal Research of the University of Oviedo PROAE with the authorisation code of 7/2018 on 23rd April 2018 (Oviedo, Asturias, Spain).

### 2.3. Adaptation Phase

Before the surgery, the 10 animals completed a two-week adaptation phase. This was mainly aimed at adapting them to the treadmill on which the 5 test animals would run during the osseointegration phase [29]. A treadmill (MC100, Fitfiu Fitness, Valls, Spain) specially adapted for this purpose was used, and it was coupled by a metal cage open at the top and the bottom to allow the rabbit to run without escaping. In the first week, the rabbits remained on the turned-off treadmill without movement for 10 min a day. In the second week, they remained on the moving treadmill for 10 min at the minimum speed (10 m/s) to allow them to gradually adapt to the treadmill before surgery and reduce post-operative stress.

### 2.4. Surgical Procedure

The rabbits underwent general anaesthesia for the surgical procedures; an intramuscular mixture of dexmedetomidine (Dexdomitor; Ecuphar, Barcelona, Spain), ketamine (Ketamidor; Karizoo, Barcelona, Spain) and buprenorphine (Bupaq; Richter Farma, Wells, Austria) was used for induction.

Anaesthetic maintenance was performed by inhalation with isoflurane (Isoflo, Zoetis, Madrid, Spain) at a fixed concentration of 1–2%. In addition, articaine hydrochloride with epinephrine 40/0.01 mg/m was administered via an infiltrative route in the operated area (Ultracain; Normon Laboratories, Madrid, Spain).

A single incision was made on the inside of each tibia in all animals. A full-thickness flap was opened, and two 3.0 × 8 mm Klockner Vega implants (Soadco, Escaldes Gordany, Andorra) were placed in the medial portion of each tibia in the area closest to the epiphysis. The implants were placed according to the full-drilling protocol recommended by the manufacturer, with bicortical anchorage and a distance of 6 mm was left between both implants (Figure 1A,B). The total implant sample size was 40 (*n* = 40).

### 2.5. Primary Stability Measurements

Primary stability was measured by two methods:

Insertion torque, expressed in Ncm, using the torque wrench of the Klockner implant surgical system (Soadco, Escaldes-Engordany, Andorra).

Resonance frequency analysis, expressed in ISQ, using a Penguin RFA device (Integration Diagnostics Sweden AB, Gothenburg, Sweden) and multi-peg measuring probes specific for 3-mm Klockner VEGA implants (Figure 1C).

### 2.6. Healing Caps

A 2-mm high healing abutment with laser markings after every 0.5 mm was manufactured specifically for this study (Soadco, Escaldes Gordany, Andorra) (Figure 2). This allowed the quantification of the potential vertical bone growth during osseointegration and the comparison of the groups. After placing the implants, the healing caps were placed and left submerged. A flat suture was made with simple stitches using 90% glycolide and 10% L-lactide 4/0 resorbable suture (Vicryl 4/0 Ethicon, Johnson & Johnson International; Somerville, NJ, USA) to facilitate adequate primary wound closure (Figure 1D).

### 2.7. Randomisation of Samples

Immediately after surgery, the 10 rabbits were allocated to the test group (Group A) or control group (Group B) using permuted blocks randomisation (5 blocks of 2 elements). This random assignment was carried out by an independent observer who was unaware of the objectives of the study.

### 2.8. Biomechanical Stimulus

From the day after surgery, the 5 rabbits assigned to the test group (Group A) started performing daily physical exercise on the treadmill for 6 weeks, while the osseointegration was ongoing (Figure 3). The run lasted 5 min a day initially before progressively increasing to two daily sessions of 10 min during the last 2 weeks. In the meantime, the 5 animals from the control group (group B) were kept at rest in their cages. During the 6 weeks, the animals in both groups followed the standard housing protocol, with ad libitum feeding and 12 h day/night light cycles.

### 2.9. Sacrifice

The rabbits were sedated with dexmedetomidine (Dexdomitor; Ecuphar, Barcelona, Spain) and diazepam (Valium, Roche Farma, Madrid, Spain) six weeks after the implant placement. They were sacrificed by intravenous injection of sodium pentobarbital at 100 mg/kg.

Subsequently, a full-thickness flap was created in each tibia for implant exposure, and photos were taken for the quantitative analysis of the vertical bone growth above the implants. As bone growth was not homogeneous around each implant, two measurements were taken: one at the point of minimum growth and another at the point of maximum growth. Next, the healing caps were removed, and the implant stability was measured again using Penguin RFA. Following this, the tibiae were cut, and the bone blocks were obtained with the implants included for the histological study.

### 2.10. Histological Processing

The samples were dehydrated by immersion in ethanol and embedded, without decalcification, in liquid methyl methacrylate for 15 days at 4 °C with agitation. Following this, they were kept at 32 °C for 6 days. Once polymerised, the samples were sectioned using a low-speed microtome with a diamond disc (Isomet, Bueher^®^ lake bluff, Dusseldorf, Germany) parallel to the longitudinal axis of the titanium implant to obtain central sections. These sections were stained using Stevenel’s Blue histological technique, in which the mineralised bone was stained red. An optical microscope (Nikon Eclipse 90i, Nikon Corporation, Tokyo, Japan) was used for the histological assessment of the obtained sections.

### 2.11. Histomorphometrical Analysis

The histomorphometric analysis was performed by an independent observer who was unaware of the case/control assignment using MetaMorph Meta Imaging Series 6.1 (Molecular Devices, CA, USA). The following parameters were analysed:-% BIC: percentage of bone-to-implant contact in the region of interest (ROI). The BIC was analysed exclusively in the coronal area of the implant, specifically in the area surrounding its neck. The apical anchorage zone was not included given that its relationship with the implant was more heterogeneous, and, unlike the coronal area, it was impossible to control during surgery.-New bone formation (mm^2^): the amount of newformed bone on the surface of the implant.-% of immature bone matrix: the proportion of the area of newformed bone that is immature, with disorganised collagen fibres.

### 2.12. Statistical Analysis

The continuous variables were described using the mean, standard deviation and the number of cases. After confirming their lack of normality using the Anderson–Darling test, the statistically significant differences between the group medians were determined using the non-parametric Kruskal–Wallis test. Finally, the linear correlation between the variables was measured using Pearson’s correlation coefficient.

## 3. Results

All the implant sites healed uneventfully, and all the implants were correctly osseointegrated, resulting in a 100% survival rate.

### 3.1. Vertical Bone Growth

Several implants showed vertical bone growth above the implant platform. However, this growth was uneven in both groups (Figure 3). The vertical bone growth was 1.26 ± 0.48 mm in the test implants (*n* = 20) and 0.32 ± 0.47 mm in the control implants (*n* = 20); the difference was statistically significant (*p* < 0.001; Table 1).

### 3.2. Implant Stability

Table 2 shows the primary stability values measured by insertion torque and RFA on the day of implant placement and the secondary stability ISQ values measured on the day of sacrifice after 6 weeks.

There were no statistically significant differences between the primary stability values of both groups based on the insertion torque (group A 27.00 ± 10.13 N/cm; group B 23.0 ± 10.05 N/cm; *p* = 0.279) or the ISQ values (group A 71.50 ± 3.09 ISQ; group B 72.53 ± 3.67 ISQ; *p* = 0.262). Moreover, there was no statistically significant correlation between the insertion torque values and the initial ISQ values (Pearson’s Coefficient = 0.00). There was a statistically significant increase in the stability of the implants in both the test and control groups during the osseointegration; the ISQ values of all the implants were higher at the time of sacrifice than at the time of placement (group A 82.75 ± 4.91 ISQ; group B 78.33 ± 3.68 ISQ; *p* = 0.008). However, this increase was statistically significantly higher in the test implant group than in the control implant group (Group A 11.25 ± 6.10 ISQ, 15.73%; Group B 5.80 ± 5.97 ISQ, 7.99%; *p* = 0.006).

### 3.3. Bone to Implant Contact

As shown in Table 2, the mean BIC values were higher in group A (25.14 ± 5.24%) than in group B (18.87 ± 4.45%), and the differences were statistically significant (*p* < 0.01).

On the other hand, the statistical analysis demonstrated that there is a direct correlation between the final ISQ and the BIC values, although without a strong association (Pearson’s Coefficient = 0.528)

### 3.4. Bone Neoformation

Table 2 shows the results of the measurement of the area occupied by newformed bone on the surface of each implant, as well as the percentage of immature bone matrix.

The mean area of bone neoformation was 280.50 ± 125.40 mm^2^ in group A and 228.00 ± 141.40 mm^2^ in group B. Although the differences between both groups were not statistically significant (*p* = 0.121), there was a trend of greater bone formation around the implants in the test group than in the control group. This showed a direct, although weak, correlation between bone neoformation and BIC (Pearson’s Coefficient = 0.320). Furthermore, the percentage of immature bone matrix was significantly lower in group A (15.38 ± 8.84%) than in group B (20.68 ± 9.53%) (*p* = 0.108), and it showed an inverse statistical correlation with BIC (Pearson’s Coefficient = −0.498). These findings indicate a lower number of immature collagen fibres in the loaded implants and faster bone maturation around the implants in the test group than in the control group. The results of the analysis of the stability and the histomorphometric variables above are shown in Figure 4.

### 3.5. Bone Disposition

The images of the histological sections obtained by microscopy (10×) showed a different orientation of the peri-implant bone matrix in the cases and controls (Figure 5A,B).

The calcified collagen fibres in the test implants were predominantly arranged perpendicular to the longitudinal axis of the implant on the crest module and at the level of the microgrooves (Figure 5C). In the controls, the fibres were arranged parallel to the implant (Figure 5D) and the newformed bone showed changes in the matrix deposits. These deposits were more disorganised, and they had darker aggregates that ended on the bone surface with a trabecular arrangement involving more vascular cavities of smaller size. The osteocytes were also surrounded by more immature spaces.

## 4. Discussion

The present study investigated the histological and histomorphometric findings of the bone around dental implants placed in rabbit tibiae and subjected to mechanical loading, as well as their biological stability during osseointegration. Rabbits were used for the experimental models since there was extensive literature on them, the authors had experience handling them and they are the smallest experimental animals that allow the placement of human dental implants. According to Sedlin and co-workers, humans and other mammals present similar biological responses to biomechanical stimuli, and the rabbit cortical bone remodels three times faster than that of humans. Therefore, the experimental period of 6 weeks used in this study would correspond to a period of 18 weeks in the human body [30].

It should be noted that the methodology of the study, involving the use of a treadmill to expose the animals to daily physical exercise, has been described in previously published studies [29,31,32,33]. It should be recognised that the loading model of this experimental setting may differ on the masticatory biomechanics, both in magnitude and the direction of forces and the mechanisms of transfer of the force to the peri-implant marginal bone. Despite the differences, the run on the treadmill of the test group can be biomechanically described as dynamic and based on successive impact loads, in the same way some authors describe mastication [34]. The loading time of up to 20 min per day was similar to the daily load time of the masticatory forces, which some authors consider to be 20–30 min [35]. In summary, the objective for the model of this study was to produce a biomechanical stimulus to be exerted on the bone surrounding the implants, similar to that induced by mastication and exerted on the maxillary bones and the implants they contain.

The loading model used in this study differs from others in which static loading was used. [36,37]. This is relevant since some studies have shown that the formation of peri-implant bone is only influenced by dynamic loads and not static loads [38,39].

Furthermore, a 2014 study by JingYun et al. demonstrated that the implant did not respond to a design identical to that of humans, but it incorporated the loading device and created a direct exit for the implant through the apical cortex, in addition to a bicortical anchorage. This placement of the implant can substantially modify the load transfer between the implant and the bone [37].

On the other hand, the study by Duyck et al. placed 40 implants in the tibiae of 8 rabbits [40]. In some of these implants, the authors applied dynamic load cycles via a transverse force of 14.7 N at a frequency of 1 Hz for 2 weeks. They applied 90 load cycles per day in the first week and 270 cycles per day in the second week. However, it seems clear that the direction of the applied load, which was completely perpendicular to the implant axis, is not related to the direction of the occlusal forces that transmit the load on the implants during mastication. This is an important problem as the compression forces are anabolic, whereas the torsional and shear deformations have no effect [41] or are negatively correlated with bone growth toward porous structures [42].

Furthermore, the number of load cycles per day does not correspond to the number of masticatory cycles of an average patient [43,44,45]. Therefore, the 2 weeks of study could be insufficient for complete osseointegration as it would be equivalent to 6 weeks in humans. These reasons may explain why, unlike the results of our study, the authors found no significant differences between the loaded test groups and unloaded control groups.

Finally, in other studies, the authors applied a dynamic load on the implants for a period of 3 weeks, but only twice a week. This does not correctly reproduce the load times for which the implants are exposed during mastication. As reported by Duyck, this can fall below the stimulation threshold. Furthermore, these load cycles were applied after the osseointegration, which mimics a delayed loading protocol and not an immediate loading as applied in our study [46,47,48].

The first finding of this study, following the analysis of the results, was the confirmation of a greater bone overgrowth for the implants that had received biomechanical stimuli. This overgrowth was caused by the detachment of the periosteum and its subsequent replacement over the healing caps; it helped to maintain the created subperiosteal space and allowed it to be filled with a blood clot, which is similar to the “Tent Pole” bone regeneration technique [49]. However, there were statistically significant differences between the amounts of bone overgrowth in the two groups, which indicates that the biomechanical stimuli caused greater bone remodelling with intense osteoblastic activity in the implants of the test group. This resulted in the formation of a greater volume of bone. These results support Pauwels’ theory, which in 1965 argued that bone is capable of adjustments to increasing or decreasing mechanical stimuli by adapting the volumetric dimensions of its architecture [13]. Our results are similar to those of the study by Wiskott et al. that used a similar model but applied direct and dynamic load cycles on the implants after osseointegration [13]. This study has similar limitations to those previously mentioned for the study by Duyck et al., namely the transversal application of force and the insufficient number of cycles. However, the authors observed that there was a statistically significant increase in the cortical bone volume in the inter-implant region following the application of the load. According to the authors, the response of the cortical bone to the applied stress consisted of the rearrangement of its structure and an increase in its mass [13]. On the other hand, Liu et al. placed 107 implants in 45 patients and observed a statistically significant bone gain 12, 24 and 36 months after loading [50].

The results of the present study also show a statistically significant increase in ISQ implant stability values, which were measured by resonance frequency analysis during osseointegration.

However, this increase was higher in the implants of the test group, which demonstrates the effect of biomechanical stimuli on the biological stability of the dental implants. The results are consistent with those obtained by our research group in a previously published retrospective clinical study [3]. That study showed that implant loading promotes implant stability measured by RFA. With a total sample size of 93, the measurement of the stability of rehabilitated implants with immediate loading (*n* = 28) and delayed loading (*n* = 65) showed a significantly higher increase in ISQ values the former than in the latter. Moreover, it was also observed that the greatest increase in stability occurred during the osseointegration (first weeks) in the immediate loading group and after prosthetic rehabilitation in the delayed loading group (after 12 weeks). Similar results were obtained by Akoglan et al. in a prospective clinical study involving 39 single implants in the posterior maxilla of 39 patients [51]. In this study, the implants rehabilitated by immediate or early loading protocols had significantly higher ISQ values during osseointegration than the delayed loading implants, in addition to the higher bone density in the peri-implant area. Other published studies also measured implant stability using RFA at prosthesis insertion as well as after 6 and/or 12 months of loading, although they did not share our objective, [52,53]. The results of these studies also show that implant stability increases after functional loading.

Regarding histomorphometry, our results showed that the biomechanical stimulation of the peri-implant bone significantly increased the percentage of contact between the bone and the implant in the test group (*p* < 0.01). This had already been extensively demonstrated in previous studies carried out in animal models, and it can be understood as osseointegration improvement under loading [9,11,12,54]. However, it should be noted that other authors did not find statistically significant differences between the loaded and unloaded implants [10,27,55].

A study by Liñares et al. placed 24 implants in the mandible of 8 minipigs to compare the histological differences between a group of rehabilitated implants with immediate loading and another group rehabilitated with delayed loading. The results indicated the absence of statistically significant differences between the BICs of both groups. Unlike in our study, all implants were loaded (immediate or delayed), and it is not possible to determine whether a hypothetical unloaded control group would have obtained lower BIC values [56].

In this study, the effect of applying loading on hard tissue formation around the implants was evident, and there were significant differences between the amounts of newformed bone on the implant surface (group A 280.50 ± 125.40; group B 228.00 ± 141.40; *p* < 0.05). This shows that the biomechanical stimulation of the peri-implant bone has a positive effect on the osseointegration process, since it activates the cellular and molecular mechanisms implicated in bone remodelling. This finding is consistent with data obtained in other histomorphometric studies carried out in different animal models [36,37].

Although the results show statistically significant higher ISQ and BIC values for the implants in the control group, a significant correlation between both values after osseointegration was not found. This is a controversial issue, but our results agree with those of previous studies carried out in animal models to elucidate this question [57,58,59,60]. In its essence, resonance frequency analysis is based on the natural frequency of vibration of a structure (in this case, the implant within the bone), and it is related to the stiffness of the peri-implant marginal bone, inversely correlating with the micromovement during loading [61]. A higher BIC does not necessarily represent greater rigidity or a correct orientation of the bone matrix as a function of the load, although these two factors are critical for the RFA results [62].

On the other hand, the histomorphometric analysis also revealed that the percentage of immature bone matrix around the group A implants was significantly lower than that of the group B implants (Group A 15.38 ± 8.84%, Group B 20.68 ± 9.53%, *p* < 0.05). This finding is consistent with the higher percentage of BIC in loaded implants since there is a greater cellular activity for bone matrix apposition on the implant surface.

Finally, the loaded bone showed a more organised arrangement of the collagen fibres, which demonstrated faster bone maturation caused by biomechanical stimuli. Interestingly, several implants in the test group showed areas with bone bridges that formed from the apical cortex towards the apex of the implant (Figure 6). This phenomenon has already been described by Trindade et al. and Wiskott et al. following the coincidental finding in other studies carried out in rabbit tibiae [13,63]. According to these authors, such bridge-like entities did not only facilitate a better and faster osseointegration, but they also provided a further pillar to stabilise the implant in situ [63]. In addition, although the analysis of the microscopic arrangement of the newformed matrix is exclusively qualitative, there was a trend of a perpendicular arrangement of the collagen fibres to the implant in the test group. This makes sense from a biomechanical point of view, given the near-perpendicular position with respect to the floor of the tibia of the rabbit and the perpendicular arrangement of the implants with respect to the major axis of the tibia. These findings are consistent with the outcomes of the histological analyses by Tian et al. in a recent study on the mechanoadaptive response of the alveolar bone to implant hyperloading during an experimental study in rats [64]. This concept is the main hypothesis of this study, and it gives rise to “the bone buttress theory” that was previously proposed by the authors of a previously published clinical study [3]; the osseointegration of immediately loaded dental implants can be based on the apposition of a load-oriented bone, like the flying buttresses of a Gothic church, rather than on a poorly oriented woven bone typical of osseointegration following conventional loading.

There are obvious limitations to extrapolating the results of this study carried out in rabbits to humans, for example comparing the mechanics of chewing and running, but the mechanisms of bone mechanotransduction are similar in all vertebrates. Despite these differences, the main clinical repercussion of these findings is that it opens the door to a paradigm shift concerning load times and function during osseointegration. The conventional approach requires waiting for complete integration, and it has its origin in the protocols established by Branemark, but that probably reflected prudence and not biology. The results of this study indicate that there are sufficient advantages of applying an immediate mechanical stimulus, which will probably be greater in a dentulous patient than in an edentulous one, in a one stage protocol compared to a two staged one, and in immediate loading than in a two staged or conventional load, and this can lead to an improvement in the BIC gain, higher biological stability values, and, ultimately, the possibility of aligning the elastic response of the bone with the applied stress.

## 5. Conclusions

The results of this animal study demonstrate that functional loading provides biomechanical stimuli that influence the osseointegration of implants at various levels:

Greater secondary stability of loaded implants.

Greater bone growth and greater bone-to-implant contact ratio.

Greater surface area of the newformed bone on the implant surface, with a higher percentage of mature bone matrix around it.

Better arrangement of collagen fibres and bone trabeculae for a favourable load dissipation on the supporting surface.

## Figures and Tables

**Figure 1 biology-10-00012-f001:**
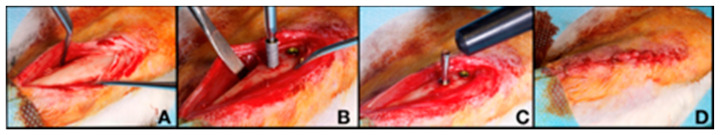
Surgical procedure. Incision and full-thickness flap (**A**). Placement of two 3.0 × 8 mm Klockner Vega implants in each tibia (**B**). Measurement of primary stability (ISQ) using a Penguin RFA device (**C**). Suture of the wound by tissue planes and primary closure over the healing caps (**D**).

**Figure 2 biology-10-00012-f002:**
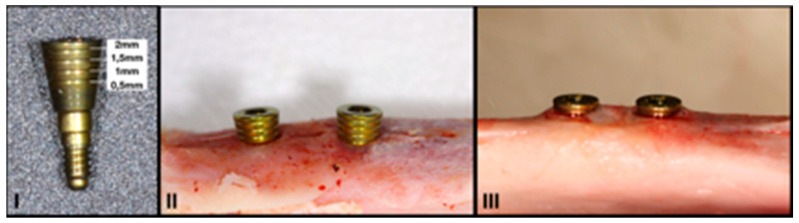
(**I**) Photo of one of the healing caps that allow for the quantification of the bone growth above the implant platform. (**II**) Image of two implants of the test group (Group A) following euthanasia. (**III**) Image of two implants of the control group (Group B) following euthanasia. It is possible to observe the differences between the heights of bone growth in the two groups.

**Figure 3 biology-10-00012-f003:**
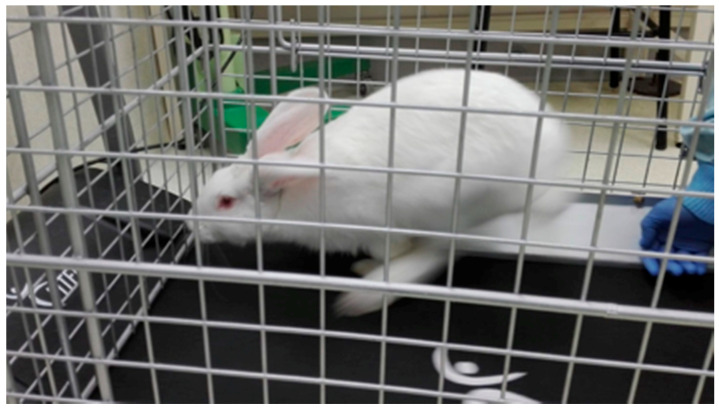
Specimen of group A exercising on the treadmill.

**Figure 4 biology-10-00012-f004:**
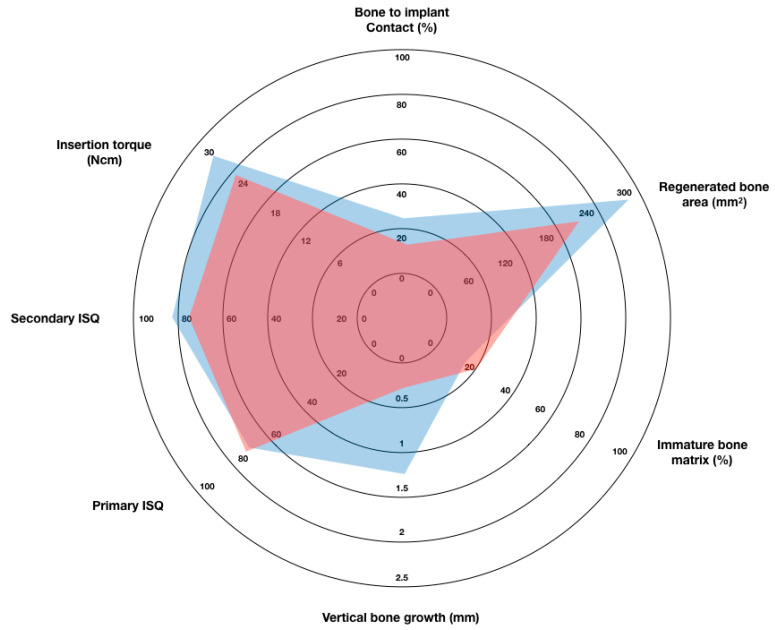
Graphic representation of the analysed stability and histomorphometric variables. Blue area represents Group A (test) and red area represents Group B (control).

**Figure 5 biology-10-00012-f005:**
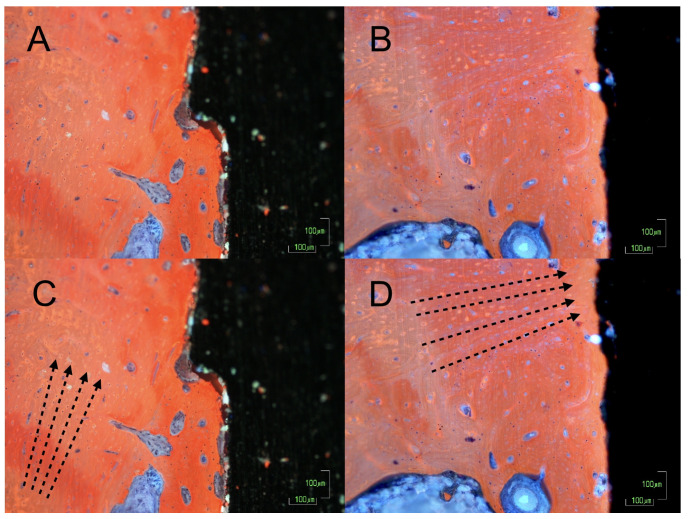
Optical microscopy (10×) image of the control (**A**) and the test (**B**). The black arrows point the parallel orientation of the matrix with respect to the implant in the control group (**C**) and the perpendicular orientation in the test group (**D**).

**Figure 6 biology-10-00012-f006:**
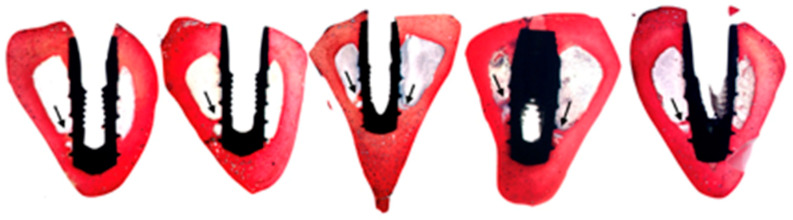
Histological sections of 5 implants in the test group (Group A) where the bone buttresses that grow from the cortex to the implant surface have been highlighted with a black arrow.

**Table 1 biology-10-00012-t001:** Table of descriptive statistics (mean and SD) of the primary stability values measured by insertion torque and RFA, as well as the secondary stability results and the increase in implant stability quotient (ISQ) values during the 6 weeks of the experiment.

Measured Parameter	Group A (Test) Mean (SD)	Group B (Control) Mean (SD)	*p* Value
Torque (N/cm)	27.00 (10.05)	23 (10.05)	*p* = 0.279
Primary stability (ISQ)	71.50 (3.09)	72.53 (3.67)	*p* = 0.262
Secondary stability (ISQ)	82.75 (4.91)	78.33 (3.68)	*p* = 0.008
ISQ increase	11.5 (6.10)	5.80 (5.97)	*p* = 0.006

**Table 2 biology-10-00012-t002:** Table of descriptive statistics (mean and SD) and statistical significance of the differences between the vertical bone growths (mm), percentages of the bone to implant contact (%), areas of regenerated bone (mm^2^) and immature bone matrix (%) of the groups.

Measured Parameter	Group A (Test) Mean (SD)	Group B (Control) Mean (SD)	*p* Value
Vertical bone growth (mm)	1.26 (0.47)	0.32 (0.46)	*p* ≤ 0.01
Bone to implant contact (%)	25.14 (5.24)	18.87 (4.45)	*p* ≤ 0.01
Areas of regenerated bone (%)	280.50 (125.40)	228.00 (141.40)	*p* = 0.121
Immature bone matrix (%)	15.38 (8.84)	20.68 (9.53)	*p* ≤ 0.01

## Data Availability

Data available in a publicly accessible repository that does not issue DOIs.

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
