# Peer review of "The Bone Buttress Theory: The Effect of the Mechanical Loading of Bone on the Osseointegration of Dental Implants"

_biology, 2020, doi:10.3390/biology10010012_

Round 1

Reviewer 1 Report

Interesting study on the use of dynamic forces to evaluate the osseointegration around implants in a rabbit model.

Although I have no concerns with the conduct of animal experiment, my main concern is with the basic assumption that the authors have made in this study related to dynamic loading of implants. The authors are comparing immediate loading in a clinical situation with the conditions described in the study (rabbits running on a treadmill) which is not correct according to me. I don't think that we can compare immediate implant loading to the functional stresses induced in the bone with a rabbit running on a treadmill. Given that there was no direct load on the implant from an occlusal aspect, one cannot correlate the findings to a clinical scenario.

Author Response

Thanks to Reviewer 1 for your valuable feedback and for considering our work as a very well tought out a executed research effort.

Like Reviewer 2, he thinks that the implants in the model used (femur-running) cannot be considered functionally loaded, taking into consideration that a “functional load would be a load that was either masticatory or simulated masticatory function on the implant not on the bone ”.

In our understanding, when a complex prosthesis - implant - bone is subjected to load, by means of functional or parafunctional forces, the stress will be transferred to all the components of this system, obviously including the bone (Skalak R. Biomechanical considerations in oseeointegrated prostheses. J Protst Dent 1983; 49: 843-8).

Our group, during the last 10 years, has extensively explored the aspects involved in the transfer of tension to the supporting bone, checking different variables:

Relating to the prosthesis:

  • Alvarez-Arenal A, Gonzalez-Gonzalez I, deLlanos-Lanchares H, Martin-Fernandez E, Brizuela-Velasco A, Ellacuria-Echebarria J. Effect of implant- and occlusal load location on stress distribution in Locator attachments of mandibular overdenture. A finite element study. J Adv Prosthodont. 2017 Oct;9(5):371-380. doi: 10.4047/jap.2017.9.5.371. Epub 2017 Oct 16. PMID: 29142645; PMCID: PMC5673614.
  • Alvarez-Arenal A, Brizuela-Velasco A, DeLlanos-Lanchares H, Gonzalez-Gonzalez I. Should oral implants be splinted in a mandibular implant-supported fixedcomplete denture? A 3-dimensional-model finite element analysis. J Prosthet Dent. 2014 Sep;112(3):508-14)
  • Martin-Fernandez E, Gonzalez-Gonzalez I, deLlanos-Lanchares H, Mauvezin-Quevedo MA, Brizuela-Velasco A, Alvarez-Arenal A. Mandibular Flexure and Peri-Implant Bone Stress Distribution on an Implant-Supported Fixed Full-Arch Mandibular Prosthesis: 3D Finite Element Analysis. Biomed Res Int. 2018 Apr 1;2018:8241313. doi: 10.1155/2018/8241313. PMID: 29805978; PMCID: PMC5899843.

Relating to the properties of the materials involved:

  • Brizuela-Velasco A, Pérez-Pevida E, Jiménez-Garrudo A, Gil-Mur FJ, Manero JM, Punset-Fuste M, Chávarri-Prado D, Diéguez-Pereira M, Monticelli F. Mechanical Characterisation and Biomechanical and Biological Behaviours of Ti-Zr Binary- Alloy Dental Implants. Biomed Res Int. 2017;2017:2785863
  • Pérez-Pevida E, Brizuela-Velasco A, Chávarri-Prado D, Jiménez-Garrudo A, Sánchez-Lasheras F, Solaberrieta-Méndez E, Diéguez-Pereira M, Fernández-González FJ, Dehesa-Ibarra B, Monticelli F. Biomechanical Consequences of the Elastic Properties of Dental Implant Alloys on the Supporting Bone: Finite Element Analysis. Biomed Res Int. 2016;2016:1850401
  • Brizuela A, Herrero-Climent M, Rios-Carrasco E, Rios-Santos JV, Pérez RA, Manero JM, Gil Mur J. Influence of the Elastic Modulus on the Osseointegration of Dental Implants. Materials (Basel). 2019 Mar 25;12(6):980.)
  • Dieguez-Pereira M, Brizuela-Velasco A, Chavarri-Prado D, Perez-Pevida E, deLlanos-Lanchares H, Alvarez-Arenal A. The Utility of Implant-Supported Fixed Dental Prosthesis Material for Implant Micromovement and Peri-implant Bone Microstrain: A Study in Rabbit Tibia. Int J Oral Maxillofac Implants. 2020 Nov/Dec;35(6):1132-1140. doi: 10.11607/jomi.8094. PMID: 33270053.

Or relating to aspects related to the surgical or rehabilitative design or of the implants themselves

  • Alvarez-Arenal A, Gonzalez-Gonzalez I, deLlanos-Lanchares H, Brizuela-Velasco A, Martin-Fernandez E, Ellacuria-Echebarria J. Influence of Implant Positions and Occlusal Forces on Peri-Implant Bone Stress in Mandibular Two-Implant Overdentures: A 3-Dimensional Finite Element Analysis. J Oral Implantol. 2017 Dec;43(6):419-428. doi: 10.1563/aaid-joi-D-17-00170. Epub 2017 Oct 3. PMID: 28972823.
  • Oliveira H, Brizuela Velasco A, Ríos-Santos JV, Sánchez Lasheras F, Lemos BF, Gil FJ, Carvalho A, Herrero-Climent M. Effect of Different Implant Designs on Strain and Stress Distribution under Non-Axial Loading: A Three-Dimensional Finite Element Analysis. Int J Environ Res Public Health. 2020 Jul 1;17(13):4738.)
  • Brizuela-Velasco A, Chávarri-Prado D. The functional loading of implants increases their stability: A retrospective clinical study. Clin Implant Dent Relat Res. 2019 Feb;21(1):122-129. doi: 10.1111/cid.12702. Epub 2018 Dec 13. PMID: 30548792.

And based on this research experience, I believe that we can agree to launch two major consensus conclusions:

- The supporting bone will be stressed in relation to the applied force and also related to the prosthesis and implant aspects.

- In this system, only one biological response should be expected: that of the bone. Regarding any of the other elements involved (protheses, implants..), an absence of mechanical failure should be required, but obviously not an adaptive response.

Taking this into consideration, the present study tries to evaluate the adaptive response of the bone during a primary healing process around an implant (an osseointegration process) when the supporting bone that contains it, it is subjected to load. During the race the femur is not only subjected to compression or tension, but also the relative curvature that is generated can also control the remodeling process (Martin B. Structure, function and adaptation of compact bone. Raven Press: New York. 1989 ).

Our objective was not to equate a masticatory load with a running load: our work attempts to describe what to expect regarding bone apposition on a titanium surface, when loads are applied to the supporting bone. And although during the article we have related them in various aspects (both are dynamic impact loads), it is clear that we have the fact that the Forces are different: both in magnitude and direction (in the masticatory a main vector follows parallel to the longitudinal axis of the implant and in our running design it will be perpendicular to that axis). And we also understand that being different they will generate a healing pattern that in one case will be efficient for chewing and in another for running. Indeed and as the Reviewer 1 argues, in our model a direct loading of the implant from the occlusal aspect cannot be described. But again the trial aims to evaluate the consequences in the bone not in the implant and we consider that if between both groups / control and trial) there is only the difference of the race, it seems clear that the results support the conclusions.

Regarding the denomination of force as "functional", we do not know how to interpret it. We have assumed that running is functional for a rabbit. If our work belonged to the field of trauma, everything would be more evident. We would be evaluating how the bone would behave during healing around two titanium fixations, when one group is subjected to exercise (functional load) and another is kept in bed. However, following the reviewers' instructions and to avoid the confusion that this association may create, we have rewritten and adapted those parts of the manuscript (including the title) in which we had made it.

Reviewer 2 Report

This a very well thought out a executed research effort.  For the most part, the manuscript is well written and presented.    However, the title, conclusions, and discussion are not supported by the research.  This focuses on the concept that implants in the femur of a rabbit were "functionally loaded".  The load applied was not a functional load on an implant as is universally envisioned.  A functional load would be a load that was either masticatory or simulated masticatory function on the implant not on the bone.  Running a rabbit on a treadmill, while interesting and innovative, does not constitute "functional loading of an implant".  If this error is not corrected, I recommend rejection of this manuscript.

I would consider this manuscript marginally acceptable if the words "non-masticatory functional loading" was placed in the title and at all points in the manuscript which now contains the words "functional loading".

I feel the authors would be better served by withdrawing the manuscript and rewriting it to illustrate the facts that their research revealed i.e.  the stretching and loading of the surrounding bone will cause improvements in the healing of an implant.  This is a concept that, to my knowledge, is not in the literature.  Does this mean that an implant placed in an edentulous site surrounded by natural teeth has a better chance of osintergration and long term survival?  That is certainly hinted at by the results.  This concept could lead to further population research on success rates/survival rates of implants in human implants placed in bone under stress (functional stress of the bone not of the implant) vs. implants in an edentulous mouth.  The authors have an opportunity to develop a new concept in the biologic conceptualization of implants but instead seem to want to force their results into a concept that the research design does not support.

Author Response

Thanks to Reviewer 2 for your valuable feedback and for considering our work as a very well tought out a executed research effort.

He thinks that the implants in the model used (femur-running) cannot be considered functionally loaded, taking into consideration that a “functional load would be a load that was either masticatory or simulated masticatory function on the implant not on the bone ”.

In our understanding, when a complex prosthesis - implant - bone is subjected to load, by means of functional or parafunctional forces, the stress will be transferred to all the components of this system, obviously including the bone (Skalak R. Biomechanical considerations in oseeointegrated prostheses. J Protst Dent 1983; 49: 843-8).

Our group, during the last 10 years, has extensively explored the aspects involved in the transfer of tension to the supporting bone, checking different variables:

Relating to the prosthesis:

  • Alvarez-Arenal A, Gonzalez-Gonzalez I, deLlanos-Lanchares H, Martin-Fernandez E, Brizuela-Velasco A, Ellacuria-Echebarria J. Effect of implant- and occlusal load location on stress distribution in Locator attachments of mandibular overdenture. A finite element study. J Adv Prosthodont. 2017 Oct;9(5):371-380. doi: 10.4047/jap.2017.9.5.371. Epub 2017 Oct 16. PMID: 29142645; PMCID: PMC5673614.
  • Alvarez-Arenal A, Brizuela-Velasco A, DeLlanos-Lanchares H, Gonzalez-Gonzalez I. Should oral implants be splinted in a mandibular implant-supported fixedcomplete denture? A 3-dimensional-model finite element analysis. J Prosthet Dent. 2014 Sep;112(3):508-14)
  • Martin-Fernandez E, Gonzalez-Gonzalez I, deLlanos-Lanchares H, Mauvezin-Quevedo MA, Brizuela-Velasco A, Alvarez-Arenal A. Mandibular Flexure and Peri-Implant Bone Stress Distribution on an Implant-Supported Fixed Full-Arch Mandibular Prosthesis: 3D Finite Element Analysis. Biomed Res Int. 2018 Apr 1;2018:8241313. doi: 10.1155/2018/8241313. PMID: 29805978; PMCID: PMC5899843.

Relating to the properties of the materials involved:

  • Brizuela-Velasco A, Pérez-Pevida E, Jiménez-Garrudo A, Gil-Mur FJ, Manero JM, Punset-Fuste M, Chávarri-Prado D, Diéguez-Pereira M, Monticelli F. Mechanical Characterisation and Biomechanical and Biological Behaviours of Ti-Zr Binary- Alloy Dental Implants. Biomed Res Int. 2017;2017:2785863
  • Pérez-Pevida E, Brizuela-Velasco A, Chávarri-Prado D, Jiménez-Garrudo A, Sánchez-Lasheras F, Solaberrieta-Méndez E, Diéguez-Pereira M, Fernández-González FJ, Dehesa-Ibarra B, Monticelli F. Biomechanical Consequences of the Elastic Properties of Dental Implant Alloys on the Supporting Bone: Finite Element Analysis. Biomed Res Int. 2016;2016:1850401
  • Brizuela A, Herrero-Climent M, Rios-Carrasco E, Rios-Santos JV, Pérez RA, Manero JM, Gil Mur J. Influence of the Elastic Modulus on the Osseointegration of Dental Implants. Materials (Basel). 2019 Mar 25;12(6):980.)
  • Dieguez-Pereira M, Brizuela-Velasco A, Chavarri-Prado D, Perez-Pevida E, deLlanos-Lanchares H, Alvarez-Arenal A. The Utility of Implant-Supported Fixed Dental Prosthesis Material for Implant Micromovement and Peri-implant Bone Microstrain: A Study in Rabbit Tibia. Int J Oral Maxillofac Implants. 2020 Nov/Dec;35(6):1132-1140. doi: 10.11607/jomi.8094. PMID: 33270053.

Or relating to aspects related to the surgical or rehabilitative design or of the implants themselves

  • Alvarez-Arenal A, Gonzalez-Gonzalez I, deLlanos-Lanchares H, Brizuela-Velasco A, Martin-Fernandez E, Ellacuria-Echebarria J. Influence of Implant Positions and Occlusal Forces on Peri-Implant Bone Stress in Mandibular Two-Implant Overdentures: A 3-Dimensional Finite Element Analysis. J Oral Implantol. 2017 Dec;43(6):419-428. doi: 10.1563/aaid-joi-D-17-00170. Epub 2017 Oct 3. PMID: 28972823.
  • Oliveira H, Brizuela Velasco A, Ríos-Santos JV, Sánchez Lasheras F, Lemos BF, Gil FJ, Carvalho A, Herrero-Climent M. Effect of Different Implant Designs on Strain and Stress Distribution under Non-Axial Loading: A Three-Dimensional Finite Element Analysis. Int J Environ Res Public Health. 2020 Jul 1;17(13):4738.)
  • Brizuela-Velasco A, Chávarri-Prado D. The functional loading of implants increases their stability: A retrospective clinical study. Clin Implant Dent Relat Res. 2019 Feb;21(1):122-129. doi: 10.1111/cid.12702. Epub 2018 Dec 13. PMID: 30548792.

And based on this research experience, I believe that we can agree to launch two major consensus conclusions:

- The supporting bone will be stressed in relation to the applied force and also related to the prosthesis and implant aspects.

- In this system, only one biological response should be expected: that of the bone. Regarding any of the other elements involved (protheses, implants..), an absence of mechanical failure should be required, but obviously not an adaptive response.

Taking this into consideration, the present study tries to evaluate the adaptive response of the bone during a primary healing process around an implant (an osseointegration process) when the supporting bone that contains it, it is subjected to load. During the race the femur is not only subjected to compression or tension, but also the relative curvature that is generated can also control the remodeling process (Martin B. Structure, function and adaptation of compact bone. Raven Press: New York. 1989 ).

Our objective was not to equate a masticatory load with a running load: our work attempts to describe what to expect regarding bone apposition on a titanium surface, when loads are applied to the supporting bone. And although during the article we have related them in various aspects (both are dynamic impact loads), it is clear that we have the fact that the Forces are different: both in magnitude and direction (in the masticatory a main vector follows parallel to the longitudinal axis of the implant and in our running design it will be perpendicular to that axis). And we also understand that being different they will generate a healing pattern that in one case will be efficient for chewing and in another for running. Indeed and as the Reviewer 1 argues, in our model a direct loading of the implant from the occlusal aspect cannot be described. But again the trial aims to evaluate the consequences in the bone not in the implant and we consider that if between both groups / control and trial) there is only the difference of the race, it seems clear that the results support the conclusions.

Regarding the denomination of force as "functional", we do not know how to interpret it. We have assumed that running is functional for a rabbit. If our work belonged to the field of trauma, everything would be more evident. We would be evaluating how the bone would behave during healing around two titanium fixations, when one group is subjected to exercise (functional load) and another is kept in bed. However, following the reviewers' instructions and to avoid the confusion that this association may create, we have rewritten and adapted those parts of the manuscript (including the title) in which we had made it.

On the other hand, the suggestion of the reviewer 2 that our work has other implications, such as the differences that may exist in the course of integration in a dentate or edentulous patient, in relation to function, seems excellent to us and we have believed It is convenient to write it in the discussion, with the aim of being more explanatory and generating new lines of research.

Round 2

Reviewer 1 Report

I have two minor comments:

Introduction: Last paragraph on Page 3 needs to be rephrased. There is a mention about immediate loading on implants in the paragraph which needs to be changed according to the context of the study.

Discussion: Last paragraph on Page 15. Please rephrase "The results of this study indicate that there are sufficient advantages of.." to the 'The results of this animal study indicate....'

Reviewer 2 Report

accept